# Characterization of inexpensive metal oxide sensor performance for trace methane detection

Daniel Furuta[1], Tofigh Sayahi[2], Jinsheng Li[1], Bruce Wilson[1], Albert A. Presto[2], Jiayu Li[1,3]

[1]Department of Bioproducts and Biosystems Engineering, University of Minnesota, 1390 Eckles Ave., St. Paul, MN 55108
[2]Department of Mechanical Engineering, Carnegie Mellon University, 5000 Forbes Ave., Pittsburgh, PA 15213
[3]Department of Mechanical and Aerospace Engineering, University of Miami, 1251 Memorial Drive, Coral Gables, FL 33146

*Correspondence to*: Jiayu Li (jiayuli@miami.edu)

**Abstract.** Methane, a major contributor to climate change, is emitted by a variety of natural and anthropogenic sources. Commercially available lab-grade instruments for sensing trace methane are expensive, and previous efforts to develop inexpensive field-deployable trace methane sensors have had mixed results. Industrial and commercial metal oxide (MOx) methane sensors, which are intended for leak detection and safety monitoring, can potentially be repurposed and adapted for low-concentration sensing. As an initial step towards developing a low-cost sensing system, we characterize the performance of five off-the-shelf MOx sensors for 2-10 ppm methane detection in a laboratory setting (Figaro Engineering TGS2600, TGS2602, TGS2611-C00, TGS2611-E00, and Henan Hanwei Electronics MQ4). We identify TGS2611-C00, TGS2611-E00, and MQ4 as promising for trace methane sensing, but show that variations in ambient humidity and temperature pose a challenge for the sensors in this application.

## 1 Introduction

Methane is a major greenhouse gas of increasing importance, with a wide variety of natural and anthropogenic sources (Saunois et al., 2016). Commercially available instruments for measuring methane are expensive, limiting the spatiotemporal resolution of monitoring and leak detection (Siebenaler et al., 2016; Riddick et al., 2020a). High-resolution monitoring networks can aid in pollution mapping and source identification, for example in reducing leaks and fugitive emissions from fossil fuel production and distribution, one of the main anthropogenic sources of methane emissions (US EPA, 2017). Accordingly, researchers are investigating several options for lower-cost trace methane sensing with the goal of developing inexpensive sensor networks (Cho et al., 2022; Riddick et al., 2020a), low-cost drone-based solutions (Liu et al., 2021), and similar monitoring methods.

One approach is to repurpose off-the-shelf methane sensors designed for industrial or residential safety use. These sensors are intended for leak detection and similar applications, and are generally specified to measure methane concentrations above trace levels (>50 ppm). Atmospheric methane concentrations are currently close to 2 ppm (NOAA Global Monitoring Laboratory 2022), and are outside of such sensors' design parameters. However, the low cost and easy availability of off-the-shelf sensors supports further investigation into their use for a broader range of applications.

Off-the-shelf methane sensors are built around several different technologies; Aldhafeeri et al. (2020) summarized the working principle of several common types of methane sensors. Infrared sensors measure the narrow-band absorption of infrared light by methane, with inexpensive sensors using simplified versions of the techniques discussed by Kamieniak et al. (2015) and Shemshad et al. (2012). Pellistor sensors measure the energy released by the combustion of gas on a catalytic surface, with challenges posed by selectivity, high energy consumption, and effects from environmental conditions such as wind speed (Kamieniak et al., 2015). Metal oxide (MOx) sensors measure the change in electrical resistance of a semiconductor layer due to interactions with gas molecules and can be made inexpensively; difficulties with cross-sensitivities and lifespan can be addressed in part with different semiconductor formulations (Meixner and Lampe, 1996). Commercially available MOx sensors have shown promise for monitoring atmospheric levels of other air pollutants besides methane, including $O_3$, CO, and $NO_2$ (Piedrahita et al., 2014; Peterson et al., 2017).

Pellistor and optical sensors are available off-the-shelf from companies such as Amphenol SGX Sensortech, at prices ranging from $80 to $300 from distributors (Mouser Electronics, Inc.). A typical example, the INIR-ME-5% optical sensor, priced at $270, is specified with a lower limit of detection of 100 ppm and is designed for methane concentrations in the percent range (SGX Sensortech, 2018). MOx sensors are often less expensive still; the components we examine range in price from $5 to $20, ideal for developing very inexpensive sensor nodes. Due to their low cost and wide availability, we focus on MOx sensors in this study.

## 1.1 Previous studies

Other researchers have investigated MOx sensors for trace methane detection, with results summarized in Table 1. Eugster and Kling (2012) used a TGS2600 sensor (Figaro Engineering Inc.) to follow diurnal trends in methane concentrations in the 1.8-2 ppm range, although with an $R^2$ of only 0.19. However, Riddick et al. (2020a) found the same method did not work with a wider concentration range and developed a different model with accuracy of 0.01 ppm. Collier-Oxandale et al. (2018) also found good results with TGS2600 sensors at two different sites, with RMSE better than 0.4 ppm; however, they found it necessary to use different calibration and regression models for the two sites, and found that their models were somewhat overfitted to other gases and environmental parameters. Although TGS2600 had useful accuracy in these studies, different calibration models were required for different sites, possibly due to different sources or environmental conditions. In particular, temperature and humidity are known to affect MOx sensor response, as do cross-sensitivities to other gases (Wang et al., 2010; Huerta et al., 2016; Nabil Abdullah et al., 2020; Kim et al., 2021). It is possible that these studies had overfitting to site-specific phenomena such as interfering gases.

Van den Bossche et al. (2017) performed a laboratory calibration of a TGS2611-E00 sensor (Figaro Engineering Inc.), finding error around 1 ppm when corrected for humidity and temperature. Bastviken et al. (2020) reported similar errors with

TGS2611-E00 in a laboratory calibration and field deployment. Taguem et al. (2021) studied the response of a sensor array

65  including TGS2611, TGS2602 (Figaro Engineering Inc.) and other sensors for biogas detection, which includes methane as a component; however, their analysis did not characterize sensor response to methane alone.

Riddick et al. (2020b) found an MQ4 sensor to match their reference instrument between 100 ppm and 9 % concentrations with an $R^2$ of 0.99. Honeycutt et al. (2019) determined a lower limit of detection for MQ4 of 82 ppm. Both studies found good

70  performance in the higher concentration range, but neither study quantified sensor performance in the background to 10 ppm range.

The cited low-concentration studies used several algorithms to fit sensor response to methane levels. Eugster and Kling (2012) used a linear regression to fit a scaled sensor resistance, which was first corrected for relative humidity and temperature, to

75  methane levels. Van den Bossche et al. (2017) followed the same method. As mentioned previously, Riddick et al. (2020a) did not find this algorithm to produce optimal results, and instead developed a more complicated non-linear regression to fit their data. Finally, Collier-Oxandale et al. (2018) evaluated 11 different regressions of varying complexity with different variables, including time, time of day, temperature, and humidity, finding that different models worked better at their different sites. Models for these studies were selected for performance rather than clarity or ease of interpretation. Although it may give worse

80  performance, we believe a simpler model may better illustrate the effects of different conditions on sensor response in our comparative study.

The cited studies use different performance metrics, with some reporting differing results for the same sensors. Eugster and Kling (2012) found an unimpressive $R^2$ of 0.19 for TGS2600, for example, while Riddick et al. (2020a) found an impressive

85  accuracy of 0.01 ppm for the same sensor. As these varying results are difficult to interpret and compare, an evaluation of these inexpensive sensors with a consistent methodology will be helpful for future work in the field.

**Table 1. Summary of results of selected studies.**

| Study | Sensor | Methane concentration | Performance |
|---|---|---|---|
| Eugster and Kling (2012) | TGS2600 | 1.85 to 2 ppm | $R^2$=0.19 |
| Riddick et al. (2020a) | TGS2600 | 1.85 to 5.85 ppm | ±0.01 ppm |
| Collier-Oxandale et al. (2018) | TGS2600 | approx. 2 to 7 ppm | RMSE=0.38 ppm |
| Van den Bossche et al. (2017) | TGS2611-E00 | 2 to 9 ppm | generally 1 ppm accuracy |
| Bastviken et al. (2020) | TGS2611-E00 | 2 to 500 ppm | ±1.1 ppm at near-atmospheric levels |
| Riddick et al. (2020b) | MQ4 | 100 ppm to 9 % | $R^2$=0.99 |
| Honeycutt et al. (2019) | MQ4 | 1.85 to 995 ppm | Limit of detection=82 ppm |

Background methane levels at our location are approximately 2 ppm, consistent with the low-concentration studies in Table 1. Eugster and Kling (2012) have a small concentration range of 1.85-2 ppm, while the other low-concentration studies have upper concentrations of 5.85 ppm (Riddick et al., 2020a), 7 ppm (Collier-Oxandale et al., 2018), and 9 ppm (Van den Bossche et al., 2017). To allow comparisons with these previous studies, for our study we chose a similar concentration range of background to 10 ppm. We chose a slightly higher upper concentration with the additional goal of evaluating suitability for detecting minor leaks and similar enhancements, while focusing on the apparently difficult-to-sense low-concentration range.

### 1.2 Study goals

Previous studies suggest that some MOx sensors may be useful at ambient methane concentrations; however, as the studies use different experimental methods, algorithms, and concentration ranges, making direct comparisons between the sensors is difficult. In this study, we compare the performance of five MOx sensors for trace methane detection with the hope of clarifying the suitability of these low-cost devices for atmospheric and near-source applications. We characterize the effect of temperature and humidity on sensor response over a methane concentration range of background (2 ppm) to 10 ppm. By systematically examining the effect of environmental conditions on sensor performance and by providing a consistent methodology, we provide a direct comparison between the different sensors and an indication of their suitability for trace methane sensing use.

### 2 Methods

#### 2.1 Sensor selection and configuration

We selected five inexpensive, off-the-shelf MOx sensors to test, detailed in Table 2. Three of the sensors (TGS2611-C00, TGS2611-E00, and MQ4) are marketed specifically for methane response (Figaro, 2013; Henan Hanwei Electronics Co., n.d.); one sensor, TGS2600, is marketed for general air pollutant response (Figaro, 2021a); and one sensor, TGS2602, is marketed primarily for volatile organic compound (VOC) detection (Figaro, 2021b). We chose to include TGS2602, which is not marketed as sensing methane, to confirm whether it could be used in sensor arrays to help account for cross sensitivities, as might be the case if it showed no response to methane.

The TGS2611 models are built on the same base sensor, with the TGS2611-E00 including an additional filter to block possible interfering gases such as alcohols. The TGS2611-C00 does not include this filter and is marketed for fast response. We believe that the manufacturer has not publicly disclosed the specific material used in the filter. TGS2600 and TGS2611-E00 have been characterized for trace methane sensing by the studies listed in Table 1.

**Table 2. Sensors used in this study. Prices for the Figaro sensors and MQ4 are from the distributors Maritex Co. and Pololu Corp.,**
**respectively, and were collected in November 2021.**

| Manufacturer | Model | Approximate Price | Target gas |
|---|---|---|---|
| Figaro Engineering Inc. | TGS2600 | $15 | general air pollutants |
| Figaro Engineering Inc. | TGS2602 | $17 | VOCs, odorous gas |
| Figaro Engineering Inc. | TGS2611-C00 | $20 | $CH_4$ - fast response |
| Figaro Engineering Inc. | TGS2611-E00 | $20 | $CH_4$ - filtered |
| Henan Hanwei Electronics | MQ4 | $5 | $CH_4$ |

MOx sensors respond to gas concentrations with varying electrical resistances. To convert this to a voltage signal which our equipment could record, we placed each sensor in a voltage divider configuration with a 10 kΩ resistor. Since the sensor response to our range of conditions was unknown, we selected the resistance as a reasonable value based on manufacturer datasheets. The sensors have internal heaters, which are intended to provide the stable elevated operating temperatures required for the metal oxide elements.

Our experiment tested three replicates of each sensor for a total of 15 sensors. The Figaro sensors of each type were from the same production batch; the MQ4 sensors were ordered from the distributor at the same time but did not have printed batch information. We designed a printed circuit board (PCB) for the sensors and associated electronics, shown in Fig. 1. The PCB holds the sensors, voltage divider resistors, and connectors for sensor signals and power. The PCB also contains power supply bypass capacitors to reduce any power supply noise, which were not in the direct path of the measurement circuit. We randomized the order in which the sensors were placed on the PCB to minimize possible thermal differences between the edge and center positions. The sensor board and sensors were burned in for several days before use.

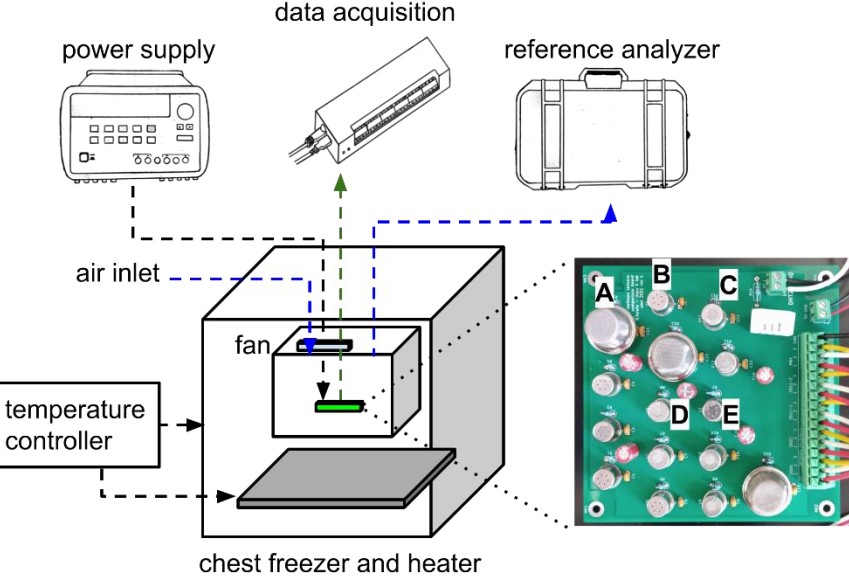

135

**Figure 1. Diagram of the experimental setup. One of each type of sensor is labelled on the circuit board: A = MQ4; B = TGS2602; C = TGS2611-E00; D = TGS2611-C00; E = TGS2600. In the actual experiment the data acquisition unit was placed inside the inner enclosure; for visual clarity it is shown outside in the diagram.**

### 2.2 Instrumentation and experimental setup

140 The sensor signals leaving the PCB were analog voltages that required digitization with a data acquisition unit (LabJack Corp. T7) connected to our host computer. The data acquisition unit has a 16-bit analog digital converter, which was further averaged internally to an effective bit depth of 19.1 bits, corresponding to a 37 μV resolution (LabJack Corp. n.d.). We used a standalone device (VWR International LLC Traceable Logger-Trac) placed next to the sensor PCB to measure and record relative humidity and temperature. The humidity and temperature logger recorded an entry every 30 seconds, which we later averaged

145 to one minute.

We powered the sensor board with a regulated bench power supply (Keysight Technologies E3631A), with the data acquisition unit powered via USB from the host computer. The sensors were warmed up for two hours prior to beginning data collection. The data acquisition unit digitized the analog sensor voltage divider signals and sent voltage readings to the host computer, as

150 directed by a program running in Python 3.7 (Python Software Foundation). The host computer paused for five seconds

between consecutive samples; including the time required to acquire data and small variations in system timing, this resulted in an entry of readings from all 15 sensors every five to six seconds, which we averaged later to one minute resolution.

We used an LI-7810 optical trace gas analyzer (LI-COR Inc.) as a reference instrument. The LI-7810 measures methane with better than 1 ppb precision and water vapor with better than 50 ppm precision, with a methane response time specified as less than two seconds (LI-COR, 2021). The reference instrument produced readings every second, which we later averaged to the minute. The analyzer was still in its initial factory calibration period.

We built our test chamber inside a commercially available chest freezer, shown in the diagram in Fig. 1. We placed a heater (Vivosun 10"×20.75" seedling heat mat) at the bottom of the freezer, with both the freezer and heater controlled by a temperature controller (Inkbird ITC-308). As our test chamber, we placed a plastic tub inside the freezer with the sensor PCB, data acquisition unit, and temperature and humidity logger enclosed. The host computer, reference instrument, and power supply were outside of the chamber, with connections made through the lids of the freezer and tub. The reference instrument continuously drew air from the test chamber by its internal sampling pump at a flow rate of 0.25 lpm, sampling from the top of the test chamber as seen in Fig. 1, and venting its exhaust to the laboratory. This sampling airflow was the only driver of air movement into or out of the chamber. Makeup air was provided from ambient laboratory air via an inlet tube, which also connected to the top of the test chamber. The sensor PCB was placed at the bottom of the test chamber, and a small fan inside the test chamber ensured mixing.

The reference instrument was connected to the test chamber by approximately four feet of tubing. We found that the instrument responded to methane pulses injected into the test chamber within seconds; as we later averaged all data to a minute time scale, we considered this lag to be negligible and did not synchronize the systems further beyond ensuring consistent system clocks.

To generate methane pulses, we drew a small portion of gas from a 99.99% methane cylinder (GASCO Affiliates, Ltd.) into a syringe and then injected the gas into the inlet tube, flushing the tube several times with ambient air. This caused a sharp spike in methane concentrations inside the test chamber, which then slowly decayed to background levels as the air was replaced with ambient air. With our chamber and reference instrument air flow, a 10 ppm concentration decayed to the background 2 ppm in around a day.

We examined sensor performance at five temperatures: 10°C, 15°C, 20°C, 25°C, and 30°C. We injected two methane pulses at each temperature for a total of ten methane decay curves over ten days. We recorded methane and water vapor concentrations with the LI-7810 reference instrument, and temperature and relative humidity with the VWR logger. Due to high ambient humidity in the laboratory, we were unable to control humidity in our test chamber.

## 2.3 Data processing and analysis

185 We averaged all collected data to a one-minute timescale, and then associated the data from the sensor PCB, reference instrument, and temperature/humidity logger by their timestamps. Since our interest is in sensor performance at background and near-source methane concentrations, we discarded data points with a reference instrument methane reading above 10 ppm.

MOx sensors convert gas concentrations into electrical resistances. Our sensor PCB converted these resistances to voltage
190 signals with a voltage divider arrangement with a 10 kΩ fixed resistor and a 5 V voltage supply. To convert a recorded sensor voltage $V_{OUT}$ to its originating sensor resistance $R_S$, we used the standard voltage divider equation Eq. 1. We used these resistances $R_S$ for our analyses.

$$R_S = \frac{(5V)(10000\Omega)}{V_{OUT}} - 10000\Omega \tag{1}$$

Due to the decaying pulse approach used, the dataset is heavily biased towards lower methane concentrations. To better evaluate sensor performance across the entire concentration range of interest, we performed our analysis by stratifying the dataset. We divided the data into three equally-spaced bins by methane concentration, spaced between the minimum recorded concentration and 10 ppm. We then randomly chose an equal number of samples from each bin, with the number chosen from each bin being equal to the size of the smallest bin, and used this sampled dataset for further analysis.

MOx sensors are sensitive to environmental conditions, particularly humidity and temperature (Abdullah et al., 2020). In our experiment, ambient temperature and water vapor concentrations were highly correlated (Pearson's r=0.95). Humidity directly affects MOx sensor response (Wang et al., 2010); the sensors we used all have internal heaters to provide elevated temperatures for their elements, and further work would be required to quantify the effect of ambient temperature on these operating
conditions. As our experimental setup was poorly suited to differentiate between the effects of temperature and humidity, we accordingly chose to only include water vapor concentration as an environmental factor in our analysis, which had slightly stronger correlation with most of the sensors than did temperature. Water vapor concentrations were measured by the LI-7810 in ppm, which we converted to percent. Determining the different effects of temperature and humidity will be an important element of future work with these sensors.

To compare the sensors' suitability for trace methane monitoring, we performed multi-variable linear regressions individually for each sensor. As our goal is to compare the performance of the different sensors in a systematic way rather than to develop a field-deployable system, we chose to use a simple, easily interpreted model rather than a higher-performing but more complex algorithm. Each regression used methane concentration measured by the reference instrument as the target. Sensor response,

water vapor concentration, and the interaction between sensor response and water vapor were predictor variables. Equation 2 shows the regression equation, in which the α and β terms are least-square regression parameters and ε is the residual error.

$$CH_4 = \alpha + \beta_1 (sensor\ resistance) + \beta_2 (H_2O) + \beta_3 (H_2O)(sensor\ resistance) + \varepsilon \qquad (2)$$

## 3 Results

Figure 2 shows the time series of the full dataset. This dataset contains 14,065 entries, representing one-minute averaged data for ten days. Gaps in the data represent periods with methane concentrations above 10 ppm due to overshooting the target concentration on pulse injections, which we removed from the dataset.

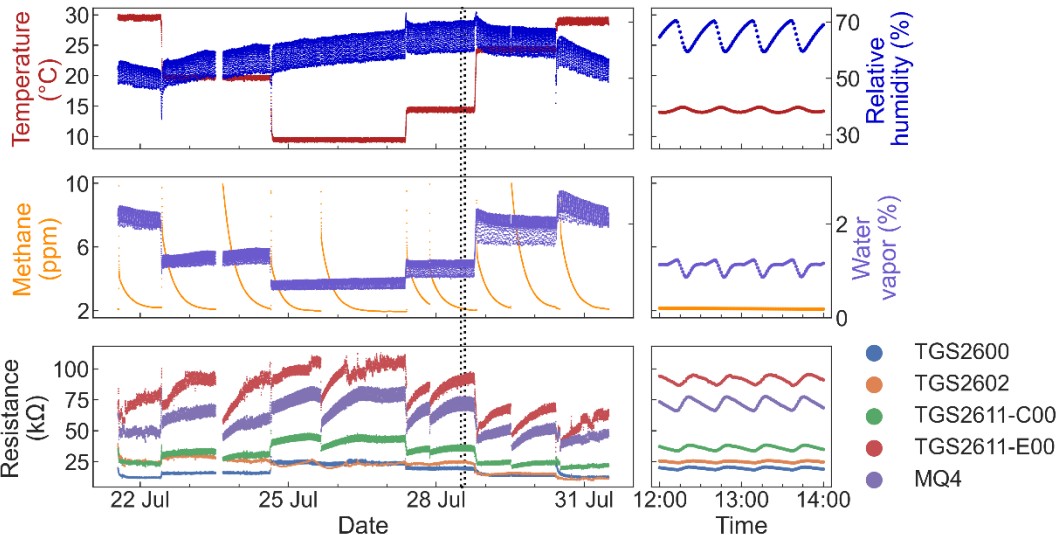

**Figure 2. Time series of the full dataset. The left plots show the entire experiment; the right plots show an expanded view of the two hours inside the dotted lines. For visual clarity, only one replicate of each sensor type is shown.**

Due to high humidity variation in the ambient laboratory environment, the experimental setup was unable to maintain stable relative humidity levels. Although this behavior was not desired, our experimental design still provided a range of humidity levels for our analysis.

As can be seen in Fig. 2, relative humidity, water vapor levels, and temperature all show a regular cycle, with levels fluctuating several times per hour. This cycle was synchronized with the on/off period of our temperature control system. From observation of the system, we believe that the cycle in water vapor levels was caused by condensation and evaporation on the walls of the test chamber due to the cooling and heating temperature cycle. The temperature fluctuations also affected relative humidity via the associated changes in vapor pressure.

Figure 3 shows the sensors' response to methane concentration and environmental parameters, including absolute humidity, temperature, and relative humidity. This figure also demonstrates the correlations among the different sensors. Most of the MOx sensors show a strong correlation with temperature and water vapor concentration ($|r| > 0.8$ for both parameters for all sensors except for TGS2602); temperature and water vapor concentrations were themselves highly correlated in our experiment. Sensor correlation with temperature and water vapor is stronger than their correlations with methane

concentrations for all of the sensors, suggesting that accounting for environmental factors is critical to using these sensors. The MQ4 and TGS2611-E00 sensors are moderately correlated with methane ($r=-0.46$ and $-0.52$, respectively). TGS2600, TGS2602, and TGS2611-C00 show low correlation with methane concentrations ($r=-0.20$, $-0.10$, and $-0.32$ respectively).

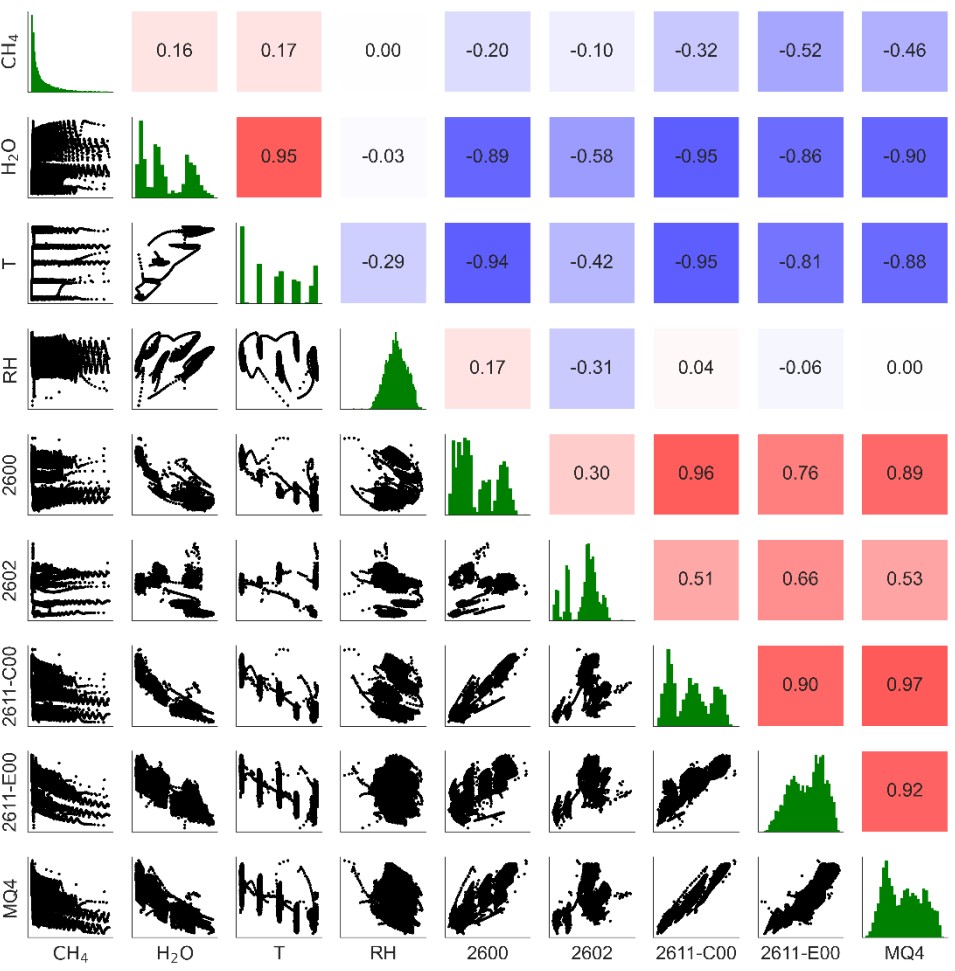

**Figure 3. The lower triangle shows pair plots between variables; the upper triangle shows correlation values (Pearson's r); the green**
**plots on the diagonal show distributions of each variable. 2600, 2602, 2611-C00, 2611-E00, and MQ4 are the MOx sensors; CH₄ and H₂O are methane and water vapor concentrations; T and RH are temperature and humidity. For clarity, only one of each type of sensor is shown.**

Consistency of measured resistance between replicates of the same type of sensor is summarized in Fig. 4. Pair and relative frequency plots are shown for each sensor type. The TGS2611-series sensors show good correlations between replicates (r=1.00). The other sensors also show good correlation (r≥0.97) but have greater variation in slopes between units, suggesting that individual, per-sensor calibration might be necessary.

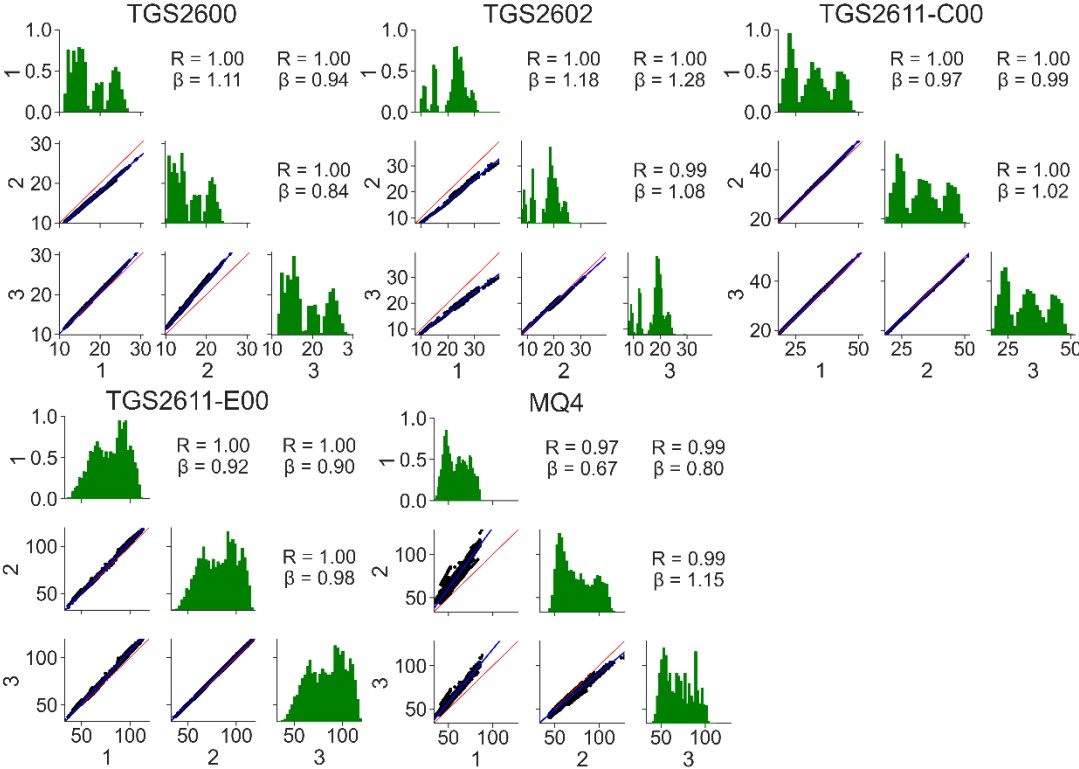

**Figure 4. The experiment contained three replicates of each sensor type; each subplot shows the relationship between the three replicates of a given sensor. The lower triangles show pair plots; the upper triangles show correlation values (Pearson's r) and the slope β of the best fit line for each pair; the green plots on the diagonal show distributions. The x-axes for all plots and the y-axes for the lower triangles are electrical resistance values in kΩ.**

A comparison of the full and stratified sampled data set is shown in Fig. 5. The stratified dataset provides a better balance among the number of observations at different concentrations. Nonetheless, low methane concentrations still had more observations. This bias could be reduced by using more bins in the sampling, at further cost to the overall dataset size.

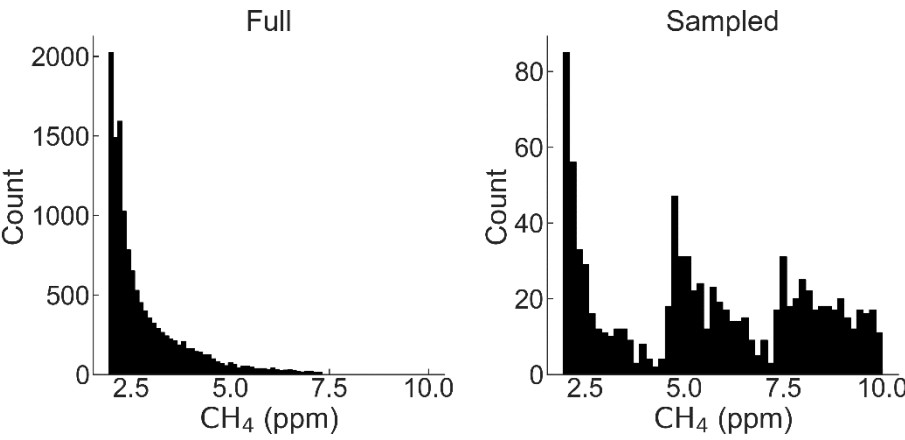

**Figure 5. Distributions of the full and sampled dataset. The full dataset has n = 14065 points. The sampled dataset has n = 930 points.**

Regression parameters and goodness-of-fit statistics for the regression model of Eq. 2 are given in Table 3. Plots of predicted versus observed methane concentrations are shown in Fig. 6. The coefficients of determination for TGS2600 and TGS2602 are poor, and there are essentially no trends between predicted and observed values for these sensors. Regression statistics for the TGS2611 sensors suggest that the regression relationships are capturing key trends in the observed methane concentrations. In general, their regression analyses tend to overpredict the low concentrations and underpredict higher concentrations. More advanced regression analysis may be successful in improving the performance of these sensors. The best regression relationships were obtained with the MQ4 and TGS2611-E00 sensors. Coefficients of determination were generally the largest, and the regression coefficients were significantly different from zero. However, the MQ4 sensors showed greater variability between units in goodness of fit than the TGS2611 sensors, with one of the three units performing worse than the TGS2611-E00 sensors. As previously mentioned, the MQ4 sensors did not have manufacturing batch information, and may have been packaged together from different batches by the component distributor. We additionally fit the regression equation with log-transformed data, but did not find improvement in the results.

**Table 3. Regression performance and coefficients for Eq. 2**

$$CH_4 = \alpha + \beta_1(sensor\ resistance) + \beta_2(H_2O) + \beta_3(H_2O)(sensor\ resistance) + \varepsilon$$

with $CH_4$ in ppm, $H_2O$ as percent, and sensor resistance in k$\Omega$. Every cell shows values for each of the three replicates of the given sensor. Statistical significance for each term is indicated with the following codes: * $p < 0.1$; ** $p < 0.05$; *** $p < 0.001$. The fit is evaluated for the 2 to 10 ppm methane range.

| Sensor | $R^2$ | RMSE (ppm) | $\alpha$ | $\beta_1$ | $\beta_2$ | $\beta_3$ |
|---|---|---|---|---|---|---|
| TGS2600 | 0.16 | 2.31 | 16.31*** | -0.60*** | -4.97*** | 0.28*** |
| | 0.16 | 2.31 | 16.39*** | -0.66*** | -4.71*** | 0.29*** |
| | 0.16 | 2.31 | 15.83*** | -0.55*** | -4.66*** | 0.26*** |
| TGS2602 | 0.06 | 2.44 | 3.34** | -0.0065 | 0.51 | 0.053 |
| | 0.06 | 2.44 | 2.66* | 0.033 | 0.92 | 0.039 |
| | 0.06 | 2.44 | 5.29*** | -0.12 | -0.51 | 0.13** |
| TGS2611-C00 | 0.67 | 1.45 | 30.86*** | -0.42*** | -0.29 | -0.33*** |
| | 0.68 | 1.43 | 30.34*** | -0.40*** | -0.30 | -0.32*** |
| | 0.68 | 1.42 | 30.76*** | -0.42*** | -0.64 | -0.31*** |
| TGS2611-E00 | 0.78 | 1.19 | 25.19*** | -0.20*** | -4.20*** | -0.0006 |
| | 0.78 | 1.18 | 24.34*** | -0.18*** | -3.75*** | -0.0076* |
| | 0.76 | 1.22 | 24.81*** | -0.18*** | -4.10*** | -0.0027 |
| MQ4 | 0.82 | 1.07 | 22.07*** | -0.18*** | 0.60* | -0.11*** |
| | 0.70 | 1.39 | 17.64*** | -0.08*** | 3.10*** | -0.13*** |
| | 0.79 | 1.16 | 20.65*** | -0.13*** | 2.09*** | -0.13*** |

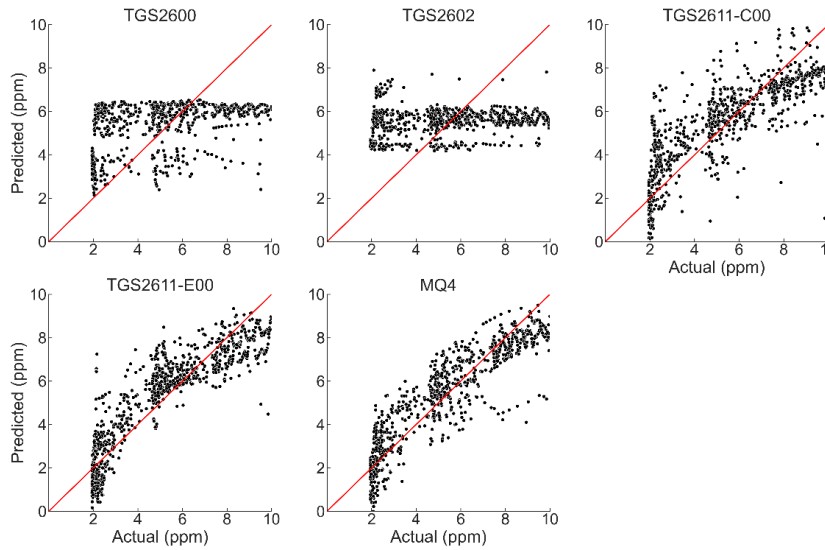

**Figure 6. Regression performance using the coefficients given in table 3, shown against the identity line. For visual clarity, only one replicate is shown for each sensor type.**

## 4 Discussion

### 4.1 Sensor performance

We tested five low-cost off-the-shelf MOx sensors to determine their suitability for monitoring methane at concentrations close to atmospheric levels. We identified TGS2611-C00 and TGS2611-E00 (Figaro Engineering Inc.), and MQ4 (Henan Hanwei Electronics Co. Ltd.) as promising candidates, with the best $R^2$ (>0.65) and RMSE (<1.5 ppm).

TGS2600 and TGS2602 sensors (Figaro Engineering Inc.) did not correlate well with methane levels in the analysis we
performed. As TGS2602 is not marketed specifically for methane, this is unsurprising. TGS2600 is marketed as a general purpose sensor, but has been used previously for methane work. Although it is possible that more complex algorithms could provide useful results, as in Eugster and Kling (2012), the inconsistent results of these algorithms in previous studies (Riddick et al., 2020a) and the availability of apparently better sensors for the same cost leaves little reason to use TGS2600 or TGS2602 for future trace methane work. However, their lack of response to methane suggests they may have potential if deployed
alongside other sensors to differentiate between methane and other gases.

Our RMSE values for TGS2611-E00 of 1.2 ppm were close to the errors of approximately 1 ppm reported by van den Bossche et al. (2017) and Bastviken et al. (2020). Our $R^2$=0.16 for TGS2600 was close to the $R^2$=0.19 reported by Eugster and Kling (2012). However, our RMSE for TGS2600 of 2.3 ppm was worse than the impressive 0.38 ppm and 0.01 ppm errors reported

by Collier-Oxendale et al. (2018) and Riddick et al. (2020a) respectively; unlike these previous studies, we did not find that TGS2600 was sensitive to methane at low levels. Possibly these studies had site-specific effects that led to better sensor performance which were not present in our experiment, or our errors could be reduced with more sophisticated models.

We found that the TGS2611 sensors had similar performance between units of the same type, but that the MQ4 units showed
greater variability. The three MQ4 sensors we tested gave $R^2$ values in our model from 0.70 to 0.82; by comparison, the three TGS2611-C00 had $R^2$ values from 0.67 to 0.68, and the TGS2611-E00 sensors from 0.76 to 0.78. The slope of the response also varied more between units for the MQ4 units than for the TGS2611 sensors. As a result, individual calibration or sensor selection is likely necessary for MQ4, adding to the cost and labor required in assembling a sensor network. In contrast, other types of inexpensive electrochemical sensors have been shown to perform well for other air pollutants without individual
calibration (Malings et al., 2019).

The 10 kΩ resistor value we used for our voltage dividers was not chosen optimally, as can be seen from the range of sensor resistances shown in Figs. 2 and 4. As the voltage divider would have the highest sensitivity when the fixed resistor is close to the sensor resistance, larger voltage divider resistors would perform better. Accordingly, for the range of conditions we studied,
resistors closer to 20, 40, and 75 kΩ would be preferable for TGS2600, TGS2611-C00, and TGS2611-E00 and MQ4 respectively.

### 4.2 Suitability for atmospheric sensing and potential applications

As background methane levels are around 2 ppm, RMSE values of greater than 1 ppm suggest that these sensors are not suitable for detecting variations in methane levels around atmospheric concentrations without more sophisticated algorithms or sensing
system designs. In particular, more precisely accounting for temperature and humidity response may improve performance. This could be achieved by regulating the temperature and humidity levels of the sampled air or by characterizing sensor response to temperature and humidity separately and developing a more complex algorithm to remove these interferences.

The level of accuracy we found from TGS2611 and MQ4 may be usable without further improvement in sensor networks for
major leak detection in urban areas, which show methane enhancements above 1 ppm in surveys, with levels sometimes exceeding 10 ppm (Phillips et al., 2013; Defratyka et al., 2021). Leaks around natural gas infrastructure (Cho et al., 2022) should also be easily detectable, with even regional-scale enhancements exceeding 1.5 ppm around production sites (Caulton et al., 2014). Due to the low cost of these sensors, such leak detection sensor networks could provide an accessible method for identifying and correcting major fugitive emission sources.

## 4.3 Sensitivity to environmental conditions

Water vapor and temperature have a large effect on sensor response, as shown by previous studies (Wang et al., 2010; Huerta et al., 2016; Abdullah et al., 2020; Kim et al., 2021). Our results show a strong correlation between absolute humidity and sensor response ($|r|>0.85$ for the TGS2611 sensors and MQ4), but little correlation with relative humidity ($|r|<0.1$ for the same sensors).

Figure 7 emphasizes the importance of temperature and humidity by showing the responses of three sensors for subsets of the data, selected to show two disjunct ranges each for temperature and water vapor. At higher temperatures the sensors have lower resistance values than at lower temperatures; the same is true for water vapor levels. In the methane concentration range we examined, the change in sensor response due to these environmental conditions can exceed the change in sensor response due to methane levels. Accordingly, field deployments of these sensors will require system designs that control temperature and humidity, co-located measurements and algorithms to account for these factors, or both.

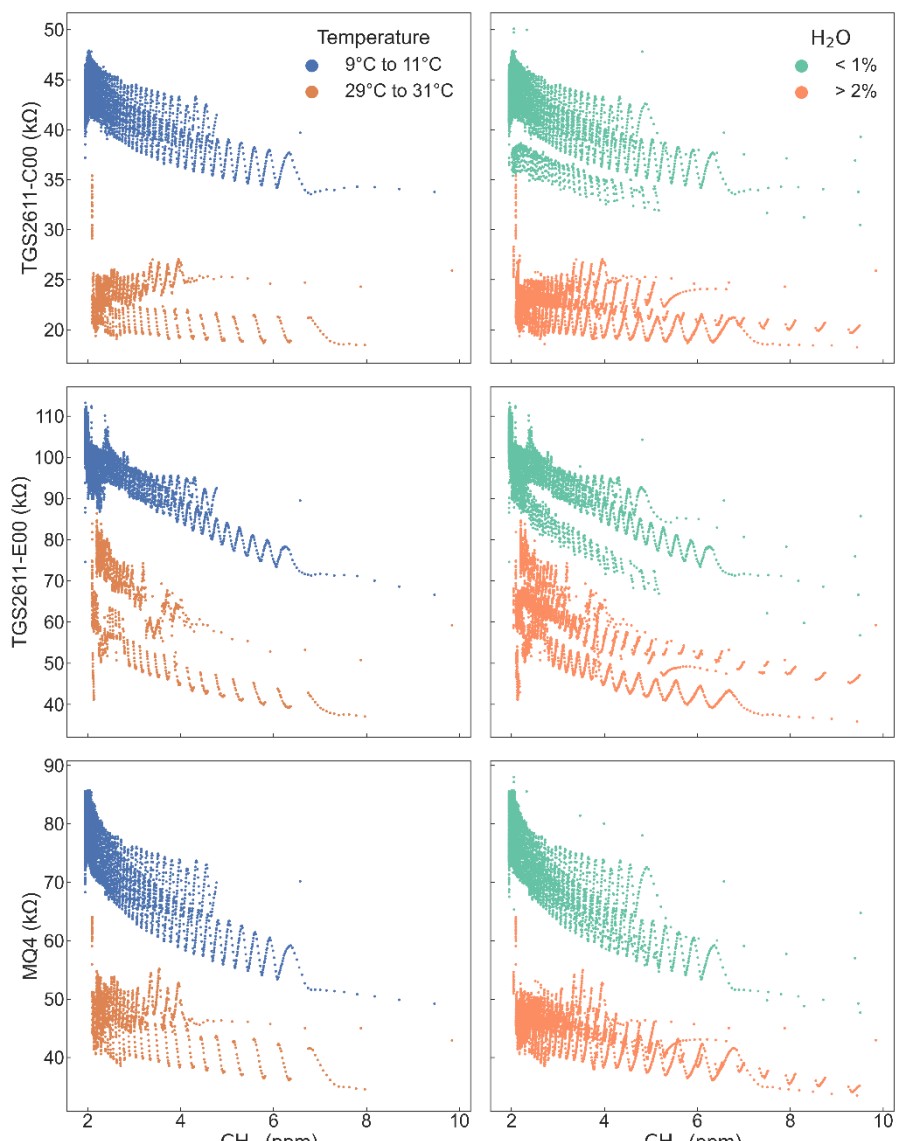

**Figure 7. Sensor response (y axis) to environmental conditions can be greater than to methane (x axis) at low concentrations.**

### 4.4 Study limitations

Our study had several limitations that will need to be addressed for future development. Most importantly, our experimental setup was unable to accurately control humidity levels independently of temperature, leaving us unable to distinguish sensor sensitivities between the two factors. We also did not test for cross-sensitivities to other gases, such as VOCs or CO, which may be important factors in real-world deployments (Collier-Oxandale et al., 2018). The lack of response from TGS2600 and TGS2602 to methane suggests they might be useful in detecting these interfering gases in a sensor array. Further development

of this idea will require characterizing the sensors' responses to common interfering gases and then developing algorithms to account for cross-sensitivities using multiple sensors.

Our calibration equation was simpler than those used by previous studies, and was chosen for clarity and ease of interpretation rather than for optimal performance. Accordingly, our expectations for sensor performance may be more pessimistic than

360 possible with more sophisticated algorithms. However, the agreement of our conclusions with some previous work supports our general findings, and we think it is unlikely that the relative performance of these sensors will be significantly different with higher-performing models.

## 5 Conclusion

Environmental conditions have a large effect on MOx sensors, dominating the sensor response at low methane levels.

Applications of these sensors for monitoring trace methane will require careful sensor calibration and algorithms to address humidity, or system designs that reduce environmental variation. We believe that addressing environmental sensitivity is the main challenge to real-world applications with the studied sensors, but that their potential to enable inexpensive sensor networks merits further development.

Monitoring background methane concentrations with the studied sensors will be difficult. It is likely that more advanced algorithms would allow improved performance over our relatively simple model, but it remains to be seen if implementation improvements will enable these sensors to perform adequately. However, we believe that the better-performing sensors (TGS2611, MQ4) have potential immediate application in leak detection networks and similar settings, and that the performance we found should be sufficient for use around fossil fuel production infrastructure, in urban leak detection, and in

other similar applications.

## Code availability

Code and circuit board design files are available by request.

## Data availability

Data is available at https://conservancy.umn.edu/handle/11299/226714

## Author contributions

Jiayu Li and AP conceived of the project; Jiayu Li and DF designed and performed the experiment; Jiayu Li, DF, TS, BW and AP analyzed the data; DF and Jinsheng Li wrote the manuscript draft; all authors reviewed and edited the manuscript.

## Competing interests

Albert Presto is an editor for Atmospheric Measurement Techniques.

## Acknowledgements

This work was funded by the US Department of Energy through the University Coalition for Fossil Energy Research (Grant S000663-USDOE).

Randi Shandroski assisted in figure preparation.

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
