# Peer review of "Characterization of inexpensive metal oxide sensor performance for trace methane detection"

_Atmospheric Measurement Techniques, 2022_

## Author Comment (AC3)

power supply

data acquisition

reference analyzer

air inlet

fan

temperature
controller

A  B  C

D  E

chest freezer and heater

Figure 1

[Figure]

Figure 3

[Figure]

Figure 4

[Figure]

Figure 5

[Figure]

Figure 7

---

## Author Response (AR1)

**Response to review, July 28th 2022**

**Characterization of inexpensive metal oxide sensor performance for trace methane detection, Furuta et al.**

**amt-2022-110**

Thank you for your insightful and valuable comments on our manuscript. We have already responded to all comments in the interactive discussion phase, and so we reproduce an edited version of the discussion below with added descriptions of the revisions we made in response. We have rearranged some of the referee comments to better collect related revisions, and we have edited some of our responses from the discussion phase for clarity and succinctness.

For visual clarity, our revisions are shown in bold text.

In addition to the revisions discussed below, we have corrected a citation on line 23 and made minor changes for grammar, clarity, and readability throughout.

We have also changed "MOx" in the title to "metal oxide" for greater clarity.

**Referee #1 discussion**

- Why did the authors select a 10-kΩ voltage divider (resistor)? As shown in Figure 2, most MOx sensors had a resistance (Rs) far greater than 10 kΩ during the experiment. From a circuit standpoint, a voltage divider with resistance close to typical Rs values would make the measurement more sensitive or accurate. A relevant question is – why 10-kΩ for all sensors?

We agree that 10 kΩ may not be an optimal resistance. As Fig. 4 illustrates, we found Rs for TGS2600 and TGS2602 in the 10 to 30 kΩ; TGS2611-C00 in the 20 to 40 kΩ range, and TGS2611-E00 and MQ4 in the 40 to 125 kΩ range. We selected the voltage divider values as a best guess for a reasonable resistance from manufacturer datasheets and a quick initial test. Interestingly and, we believe, coincidentally, the measurements with the worst sensitivity (MQ4 and TGS2611-E00) showed the best performance.

Figures 2 and 4 provides enough information to pick better resistance values for future work, as they show the expected resistance ranges for the sensors in these conditions; we will add a note to this extent in the discussion.

**We have added discussion around lines 124 and 317.**

- Figure 1: What was the inlet airflow rate? To my understanding, the LI-7810 analyzer has a sampling airflow rate of 0.25 LPM. Did this go back to the testing chamber?

As the reviewer notes, the reference instrument has a flow rate of 0.25 LPM. We had no other pump in the system, and so airflow into the test chamber was the same 0.25 LPM. The outlet of the LI-7810 was vented to the outside laboratory, causing the gradual decrease in methane concentration in the test chamber for each pulse.

- (Again) Figure 1: What was the response time of the LI-7810 analyzer, considering the length of tubing, averaging time, and so on? Did the authors synchronize the readings from MOx sensors and the LI-7810 by considering response time differences?

The reference analyzer was connected to the test chamber by a short length of small-diameter tubing. We found that the LI-7810 responded to the methane pulse injections into the test chamber within seconds. As we averaged all measurements to the minute and the pulse decay to background took place over a relatively long time scale, we considered this error negligible and did not synchronize the readings beyond aligning system clocks for the various devices.

- I would suggest the authors list the response time of LI-7810 since it is an optical gas meter. As per the instrument supplier, LI-7810 has a response time of 2 sec for CH4 between 0-2 ppm (without considering the transfer time in tubing's). This is in fact very fast for an optical gas meter. Providing such information won't hurt (but rather help) the manuscript.

We agree. The very fast response time for the instrument is another useful detail with regards to concerns about synchronization. We will add this information to the revision.

**We have added discussion to the paragraphs from lines 154 to 172.**

- (Again) Figure 1: I saw quite a few capacitors on the PCB. Please briefly explain their purpose (to avoid unnecessary confusion regarding the measurement circuit).

The capacitors in Fig. 1 (the small red and brown components seen on the PCB) were for power supply bypassing, to reduce the effect of power supply noise and to isolate the effects of any transients. As we were using a bench power supply for this experiment these components were likely unnecessary. As the reviewer suggests, the capacitors were not in the direct path of the measurement circuit.

**We added content in line 131.**

- Please provide the ADC's bit info for LabJack T7. As per the company, a LabJack T7 may use an ADC from 12 to 24-bit. The number of bits can have a large influence on the resolution of acquired data, especially for high Rs

Our setup used the default T7 settings, which correspond to an effective bit depth of 19.1 bits, and an effective resolution of 37 µV as per the company (https://labjack.com/support/datasheets/t-series/appendix-a-3-2-2-t7-noise-and-resolution). The largest Rs we observed was less than 150 kΩ; at a 150 kΩ resistance, a 37 µV change corresponds to a resistance change of around 0.013%. We believe that errors due to ADC resolution are likely insignificant compared to the uncertainty of the sensor itself, and likely even compared to overall electrical system noise.

As our measurements were also averaged to a longer time scale for analysis (from approximately one reading every five seconds to the minute scale), the actual accuracy may be somewhat better than this simple calculation.

- The authors talked about noise-free bits. Such information is useful but not usually required as the bits are related to many factors such as sampling rates. Please simply provide the ADC bit information. Based on the authors' response, I guess it could be 24-bits, which is adequate for the sensor comparison experiments.

We agree about concisely specifying bit information. The hardware ADC operates at 16 bits; the noise floor is lowered further by averaging, giving the effective bit rate we mentioned previously. The manufacturer discusses the noise and resolution at length (https://labjack.com/support/datasheets/t-series/appendix-a-3-2-2-t7-noise-and-resolution); we will simply state the 16-bit hardware depth with averaging for 37 µV nominal effective resolution in the revision, and cite the datasheet for the details.

**We added content in line 141.**

- Line 149: Vs is a bit misleading. It is the voltage drop across the 10-kΩ voltage divider instead of the sensor (Rs). I suggest the authors use a different subscript.

We agree; $V_{OUT}$ would be more suitable.

**We revised Equation 1 and the discussion at line 191.**

- Equation 2: Even though the sensors are heated, they still suffer from temperature variation, which in turn would influence the sensors' resistance. That being said, why was temperature not included in the calibration equation?

Due to limitations of our experimental setup, temperature and absolute humidity were highly correlated (Pearson's r = 0.95, see Fig. 3). Accordingly, to avoid a non-generalizable improvement in $R^2$ we chose to only include one of the two in our calibration equation, selecting absolute humidity for the stated reasons. As you correctly note, ambient temperature will also have an effect on sensor response. As we mentioned in our manuscript, these effects will need to be untangled in future work, and this should be a priority for further calibration work using these sensors.

**We added emphasis in lines 205-209.**

- Line 187: As per Figure 2, the relative humidity decreased with an increase in temperature. To me, this is related to the temperature dependency of vapor pressure. I would suggest the authors remove "likely as the result of a condensation and evaporation cycle."

We agree that this effect is responsible for some of the change in relative humidity. However, we also saw a cycle in absolute humidity levels along with temperature. We believe this was the result of some moisture condensing on and evaporating from the walls of the test chamber as the temperature cycled.

**We expanded our discussion around line 231.**

- Line 195-196: I suggest the use of r instead of R for Pearson's correlation coefficients, to avoid unnecessary confusion (R versus R2).

We agree.

**We have changed this notation throughout.**

- Figure 4: Were all the three sensors of the same model coming from the same batch of products? Different batches of sensors could differ in response factors.

The Figaro sensors of each type were taken from the same batch. The MQ4 does not appear to have a batch code printed on it; the MQ4 sensors were ordered from the distributor at the same time, but they were individually packaged and so we are unsure whether they were from the same batch. This could be a possible influence on the greater variability for the MQ4 than for the Figaro sensors. We agree that evaluating the sensors for batch-to-batch consistency can be an important addition to future work.

**We added discussion to lines 128 and 272.**

- I would suggest the authors offer a discussion about the potential application scenarios of those CH4 sensors for ambient air and source measurement towards the end of the manuscript.

Thank you for this suggestion. Other reviewers mentioned the same issue. Prior to the preprint we revised the discussion section to emphasize this important question. In particular, we believe some of these sensors may be usable as-is for natural gas leak detection in urban areas or around fossil fuel infrastructure; it is also possible that more sophisticated algorithms or system design that reduces environmental influences may allow ambient air measurement around atmospheric levels.

**In the previous revision we restructured the discussion section; in this revision we also emphasize this point in our expanded conclusion section.**

- A final thought: As per the Figaro Company, the TGS26XX sensors' response is nearly linear (Rs/R0 versus gas concentration) on a log-log graph. I just feel there could be a calibration equation better fitting the experimental data than equation 2. Here comes a question – why was general linear regression used to build the calibration curve?

We did try fitting log-transformed data, and did not find meaningful performance improvement. The manufacturer curves for most of the sensors (excepting TGS2600) lack data in the methane concentration range we examined; possibly the sensor response in this very low range (relative to design parameters) is not linear.

We agree that Equation 2 is a crude calibration equation, and we do not propose it as necessarily the best option for a sensing system using these sensors. As our goal was to compare these devices and their sensitivity to environmental conditions, we thought a simple linear model was suited to roughly evaluating sensor response to methane and humidity/temperature, while being relatively robust with a low risk of overfitting. For real-world use, a more sophisticated model (whether nonlinear regression, machine learning, or something else) might provide better performance, but we think it unlikely that a better algorithm will upend the relative performance of these sensors.

As we note in Section 4.1, our RMSE and $R^2$ values are similar to those found in some previous studies. Two previous studies found better error values for TGS2600. Riddick et al. (2020a) had uncertainty of 0.01 ppm, but only an $R^2$ of 0.23 for their best model, as compared to our $R^2$ of 0.16. Collier-Oxendale et al. (2018) also found good performance for TGS2600 in a field study with more sophisticated algorithms, but found different algorithms to perform better at their two sites - due to the complexities of real-world deployments, there may be more influence from site-specific effects or overfitting than in our relatively controlled laboratory environment. Although we agree with the reviewer that more sophisticated algorithms could perform better, we feel that the general "ballpark" agreement of our results with previously published work supports our overall conclusions.

**We have added content to discuss these points around lines 72, 274, and 358.**

**Referee #3 discussion**

- It is not very clear why the test conditions with a concentration range between 2 and 10 ppm were chosen as this is way higher than ambient but lower than serious leaks. Also at the end you discuss applicability. So one wonders why this was chosen up front?

2 ppm is the ambient concentration at our location, which is slightly higher than but close to the global average background concentration. As we were primarily interested in characterizing the sensors close to ambient levels, rather than at higher concentrations where we would expect better performance, we did not want too high of an upper bound. However, we also did not want too narrow of a range to evaluate sensor applications, as we did not expect particularly low RMSE values and as we are interested in using the sensors for leak detection and similar uses.

The previous low-concentration studies in Table 1 have upper bounds of 2, 5.85, 7, and 9 ppm. We felt that 10 ppm was in keeping with this general range of previous work, allowing us to make some comparisons with these studies, and was not too high to evaluate performance in this more difficult concentration range for the sensors.

**We added a paragraph at line 90.**

- The manuscript should have a clear conclusions section for AMT, different from the discussion section.

**We marked the final paragraph as the conclusion, and expanded it with an additional paragraph focusing on sensor usability.**

- In the discussion to other studies it is not always clear how the calibration/regression in the current study differs from other studies. This could be useful to specify.

**We added a paragraph at line 73.**

- Some figures are unclear especially figure 7. What humidity is shown as <1% and >2%? What is shown there. Other comments below.

We will clarify that this figure shows trends for subsets of the data with different temperatures and water vapor concentrations, and is intended as an illustrative tool to show the large differences in sensor response under different environmental conditions. Primarily, we intended to show that environmental conditions can have a large impact on sensor performance. As major confounding factors, these environmental parameters need to be considered when determining the limit of detection and implementation of these sensors.

**We revised the paragraph at line 341.**

- Details
- Table headers are normally on top of the tables not under

**We corrected the formatting throughout.**

- Table 1 can you use similar performance metrics? And if similar performance metrics do not exist in the papers, may be you could use this as an additional justification of your paper?

As you note, the previous studies do not use easily comparable metrics. We will further emphasize that this was a goal of our paper.

**We added content at line 83.**

- Table 2, can you please add a date on when the price data was gathered…. Please also specify what is meant with "fast response" and "filtered" (not really explained in the text).

We will specify in the caption that the prices were current in November 2021. The TGS2611-E00 has a built-in filter to reduce interference from alcohols and other gasses, while the "fast response" TGS2611-C00 does not, according to the manufacturer. As far as we have found, the manufacturer does not specify the filter material. We will clarify this in the revision.

**We have revised the Table 2 caption and the paragraph at line 113.**

- In the discussion of the poor performance of TGS 2602 (and to an extent TGS 2600) it should be made clearer that these are NOT methane sensors.. it is mentioned later but at times they just look like poor performing sensors which is a little unfair to them.

We agree. The TGS2602 is not intended to sense methane, and the TGS2600 is advertised as a general gas sensor, but has been used in several previous methane studies. We will emphasize in section 4.1 that TGS2602 is not designed for methane, and that TGS2600 is marketed as general purpose.

**We have revised the paragraph around line 294.**

- Figure 1: could you improve it so that you see where the reference analyzer samples form relative to the PCB? Also do you have some basic data such as residence time in your chamber. Not critical but would make for a better description. Also inlet air is lab air? Or zero air? I assume lab air because of the RH issue?

We agree that more detail here would be helpful. Referee #1 raised related issues.

In addition to the revisions discussed above, we will state that the analyzer draws from the top of the chamber, while the PCB sits in the bottom, and that the inlet takes in lab air. We will also emphasize that we had a small fan in the test chamber running continuously to ensure that the test chamber air was well mixed.

**We revised lines 159 to 172.**

- The RH variability is bothersome to me. The changes in RH are substantial over short periods of time. The explanation of lab air variability seems very odd as a 10% RH variability 3 times over 1 hour is just odd (fig 2). It would be useful to strengthen this discussion. That data looks like a flow or sensor issues especially seeing how regular it is more than any real room RH variability

We agree that further clarification on this point would be useful. Referee #1 touched on the same concern.

The changes in RH occur at the same frequency as the temperature control cycling, as can be seen in Figure 2. We attribute the cycling in RH to a combination of changing vapor pressure due to temperature fluctuations and, as noted in our response to referee #1, a condensation/evaporation cycling occurring due to the temperature control. As our environmental chamber was a low-cost, ad-hoc design, this difficulty in controlling humidity was one of the main challenges in our study. As we discuss, more precise control over environmental conditions in future work would be helpful in identifying the limits of performance for these sensors.

**We have revised the paragraph at line 229.**

- Table 3 and when discussing RMSE can you please remind people of the range of values this is based upon

We will add a note in the table caption that the values are for the 2 to 10 ppm range we examined.

**We revised the Table 3 caption.**

- Figure 1 : A,B,C,D and E on the PCB are hard to see.
- Figure 4 has a poorer resolution than other figures and seems to have some lines to the left? And between first and second column of panels
- Figure 3 legend: please format the formulas with subscript.
- Figure 5 and 7. please format chemical formulas as formulas (subscript 4)

**We corrected the figures.**